

# Looking for a generic inhibitor of amyloid-like fibril formation among flavone derivatives

Tomas Šneideris[1], Lina Baranauskienė[1], Jonathan G. Cannon[2], Rasa Rutkienė[3], Rolandas Meškys[3] and Vytautas Smirnovas[1]

[1] Department of Biothermodynamics and Drug Design, Vilnius University Institute of Biotechnology, Vilnius, Lithuania
[2] Department of Natural Sciences and Engineering, Middle Georgia State University, Cochran, GA, USA
[3] Department of Molecular Microbiology and Biotechnology, Vilnius University Institute of Biochemistry, Vilnius, Lithuania

## ABSTRACT

A range of diseases is associated with amyloid fibril formation. Despite different proteins being responsible for each disease, all of them share similar features including beta-sheet-rich secondary structure and fibril-like protein aggregates. A number of proteins can form amyloid-like fibrils *in vitro*, resembling structural features of disease-related amyloids. Given these generic structural properties of amyloid and amyloid-like fibrils, generic inhibitors of fibril formation would be of interest for treatment of amyloid diseases. Recently, we identified five outstanding inhibitors of insulin amyloid-like fibril formation among the pool of 265 commercially available flavone derivatives. Here we report testing of these five compounds and of epi-gallocatechine-3-gallate (EGCG) on aggregation of alpha-synuclein and beta-amyloid. We used a Thioflavin T (ThT) fluorescence assay, relying on halftimes of aggregation as the measure of inhibition. This method avoids large numbers of false positive results. Our data indicate that four of the five flavones and EGCG inhibit alpha-synuclein aggregation in a concentration-dependent manner. However none of these derivatives were able to increase halftimes of aggregation of beta-amyloid.

## INTRODUCTION

Amyloid fibril formation is related to a number of fatal neurodegenerative disorders, such as Alzheimer's and Parkinson's diseases and transmissible spongiform encephalopaties. One of the strategies for pharmacological intervention is to inhibit protein aggregation into amyloid structure (*Bieschke, 2013*). Different proteins and peptides form amyloid aggregates in case of each disease, but all aggregates share similar physicochemical and structural properties (*Chiti & Dobson, 2006*). Structurally similar aggregates, referred to as amyloid-like fibrils (*Nelson & Eisenberg, 2006*), can be formed from both disease-related and unrelated proteins and peptides *in vitro*. This led to the idea that amyloid-like structure may be a generic property of polypeptide chains (*Chiti & Dobson, 2006*). Given

Corresponding author
Vytautas Smirnovas,
vytautas@smirnovas.info

structural similarities of different amyloid and amyloid-like fibrils, there is a possibility that generic inhibitors of amyloid fibril formation may exist.

A number of small molecules were reported as inhibitors of amyloid-like fibril formation (*Doig & Derreumaux, 2015*; *Seneci, 2015*), some of them reached different phases of clinical trials, but none is approved as a drug yet (*Mangialasche et al., 2010*; *Seneci, 2015*). One of the best-known inhibitors of protein amyloid fibrillation is epi-gallocatechine-3-gallate (EGCG). There are reports, suggesting its inhibitory effect on the fibril formation of amyloid beta (Abeta) peptide and alpha-synuclein (*Ehrnhoefer et al., 2008*; *Roberts & Shorter, 2008*; *Bieschke et al., 2010*), huntingtin (*Ehrnhoefer et al., 2006*), mammalian and yeast prions (*Rambold et al., 2008*; *Roberts et al., 2009*), kappa-casein (*Hudson et al., 2009a*), transthyretin (*Ferreira et al., 2009*; *Ferreira, Saraiva & Almeida, 2011*), islet amyloid polypeptide (*Meng et al., 2010*), *Plasmodium falciparum* merozoite surface protein 2 (*Chandrashekaran et al., 2010*; *Chandrashekaran et al., 2011*), human insulin (*Wang, Dong & Sun, 2012*), hen egg white lysozyme (*Ghosh, Pandey & Dasgupta, 2013*), tau protein (*Wobst et al., 2015*), and human parathyroid hormone (*Gopalswamy et al., 2015*). The large number and variety of targets suggested that EGCG is a genuine generic inhibitor of amyloid fibril formation. Resveratrol is another compound inhibiting amyloid-like fibril formation of several proteins, including Abeta (*Feng et al., 2009*; *Ladiwala et al., 2010*), alpha-synuclein (*Herva et al., 2014*), and islet amyloid polypeptide (*Mishra et al., 2009*). A number of different flavone derivatives, including morin, quercetin, fisetin and luteolin were reported as inhibitors of Abeta fibrillation (*Ono et al., 2003*; *Akaishi et al., 2008*; *Ushikubo et al., 2012*). Luteolin, quercetin and fisetin can inhibit transthyretin aggregation (*Trivella et al., 2012*), and luteolin also inhibits fibrillation of insulin (*Malisauskas et al., 2015*). There is a report on islet amyloid polypeptide inhibition by morin (*Noor, Cao & Raleigh, 2012*). Our interest in flavones as inhibitors of amyloid-like fibril formation was especially raised by the study of *Akaishi et al. (2008)*, which suggested that inhibitory effect of flavone derivatives is dependent on the number and positions of hydroxyl group around the flavone backbone and a subsequent work of *Ushikubo et al. (2012)*, which designed a new flavone-derived inhibitor of Abeta aggregation.

One of the major problems in the detection of anti-amyloid compounds is ambiguity of the methods used for screening. A significant portion of the studies referenced relied only on changes in maximal ThT fluorescence intensity to establish inhibition of fibril formation (*Ono et al., 2003*; *Akaishi et al., 2008*; *Ushikubo et al., 2012*), sometimes leading to controversial results. For example *Ono et al. (2003)* claimed kaempferol as an inhibitor, while *Akaishi et al. (2008)* showed it to enhance Abeta fibril formation. Other studies have described how ThT fluorescence intensity can be affected by different compounds (*Foderà et al., 2008*; *Hudson et al., 2009b*; *Noormägi et al., 2012*).

Recently, we demonstrated the ability to avoid false-positives in ThT fluorescence assay-based screening by comparing halftimes of aggregation ($t_{50}$) rather than fluorescence intensities (*Malisauskas et al., 2015*). Screening of 265 flavone derivatives based on halftimes of aggregation let us to identify 5 outstanding inhibitors (Fig. 1) of insulin amyloid-like fibril formation (*Malisauskas et al., 2015*). In order to further test our screening method and

Scutellarein    Luteolin    7,8,2'-trihydroxyflavone

3,6,2',4',5'-pentahydroxyflavone    Gossypetin

**Figure 1** **The best flavone inhibitors of insulin amyloid-like fibril formation.**

identify a possible generic inhibitor of amyloid-like fibril formation among these flavone derivatives, we checked their effect on the aggregation of alpha-synuclein and Abeta.

## MATERIALS AND METHODS

Flavones were purchased from Indofine Chemical Company and EGCG was purchased from Sigma. Flavones and EGCG were dissolved at a concentration of 20 mM in dimethyl-sulfoxide (DMSO) and stored in the dark at room temperature for up to two weeks.

### Production of alpha-synuclein

*E. coli* BL-21(DE3) (Invitrogen) was used as the host strain for the over-expression of alpha-synuclein. For this purpose, cells harbouring a plasmid pRK172 were grown in a standard NB medium supplemented with 50 µg/mL ampicillin. 200 mL of medium was inoculated with 1 mL of the overnight culture and incubated at 30 °C until an $OD_{600}$ of 0.7–0.8 was reached. Protein expression was then induced by adding IPTG to a final concentration of 0.2 mM, and the incubation was continued for additional 18 h. The cells were harvested by centrifugation for 30 min at 4,000 g (4 °C), resuspended in 20 mM Tris-HCl buffer (pH 8.0), containing 0.5 M NaCl, 1 mM PMSF and 1 mM EDTA and disrupted by sonication at 22 kHz for 3 min., using 50% amplitude. To remove cellular debris, the cell lysate was centrifuged at 10,000 g for 20 min at 4 °C. After centrifugation, cellular extract was subjected to a 20 min. heat treatment using a water bath at 100 °C. Cell extract with aggregated proteins was immediately centrifuged at 10,000 g for 30 min. at 4 °C. The resulting clear supernatant was dialysed at 4 °C for 18 h against 20 mM Tris-HCl buffer (pH 8.0), containing 1 mM EDTA and 1 mM DTT (buffer A). The desalted sample was applied at a flow rate of 1 mL/min onto a 5 mL HiTrap ANX HP column (GE Healthcare, Little Chalfont, UK), previously equilibrated with buffer A. After washing with 5 column volumes of buffer A, the recombinant protein was eluted using a linear

gradient of 0–1 M NaCl in buffer A. The eluted from the column fractions were checked by SDS electrophoresis, pooled and dialyzed overnight against buffer A. The dialyzed protein solution was applied at a flow rate of 0.5 mL/min onto second ion exchange 1 mL HiTrap Q XL column (GE Healthcare) equilibrated with buffer A. After a 5 column volume wash with buffer A, alpha-synuclein was eluted over a linear gradient of 0–1 M NaCl in buffer A. The major peak eluted from the column was checked by electrophoresis, pooled and dialyzed overnight against 5 mM ammonium carbonate buffer (pH 7.6). Desalted protein samples were flash-frozen, lyophilized and stored at −20 °C until use. The homogeneity of protein was verified by SDS-PAGE. Protein concentration was determined using the Lowry method with bovine serum albumin as the standard.

## Production of abeta

The expression vector for Abeta42 in *E. coli* was described previously (*Walsh et al., 2009*; *Vignaud et al., 2013*). The recombinant Abeta peptide was expressed in *E. coli* BL-21Star[TM] (DE3) (Invitrogen, Carlsbad, California, USA) and purified similarly to a previously described method (*Walsh et al., 2009*; *Hellstrand et al., 2010*; *Vignaud et al., 2013*). The expression vector for Abeta42 peptide was transformed into $Ca^{2+}$-competent *E. coli* cells by heat shock and spread on LB agar plates containing ampicillin (100 μg/mL). Single colonies were used to inoculate 100 mL overnight cultures in LB medium with ampicillin (100 μg/mL). The next morning 1 mL of overnight culture was transferred to 400 mL of auto-inductive ZYM-5052 medium (*Studier, 2005*) containing ampicillin (100 μg/mL) and grown for 24 h. The cell suspension was centrifuged at 5,000 g and 4 °C for 15 min. The cell pellet was frozen. The frozen cell pellet from 3.6 L culture was thawed, homogenized with Potter–Elvehjem homogenizer in a total of 100 mL buffer B (10 mM Tris/HCl pH 8.0, 1 mM EDTA) and sonicated for 10 min on ice (30s/30s horn, 50% duty cycle). Cell pellet was centrifuged at 18,000 g and 4 °C for 15 min. The supernatant was removed, and the pellet was resuspended twice in 100 mL of buffer B, homogenized and centrifuged as above. The third supernatant was removed and the pellet was resuspended in 50 mL of buffer C (8 M urea, 10 mM Tris/HCl pH 8.0, 1 mM EDTA), homogenized and centrifuged as above, resulting in clear solution. The urea-solubilized inclusion bodies (50 mL) were diluted with 150 mL of buffer C, added to 50 mL DEAE-sepharose equilibrated in buffer C, and gently agitated (80 rpm) for 30 min at 4 °C. The slurry was then applied to a Büchner funnel with Fisherbrand glass microfibers paper on a vacuum glass bottle. Later, the resin was washed with 50 mL of buffer C and then with 50 mL of buffer C with 25 mM NaCl followed by four aliquots of buffer C with 125 mM NaCl. Each aliquot was incubated with the resin for 5 min before collection under vacuum. The presence of the peptide in aliquots was tested using Tricine SDS-PAGE (*Schägger, 2006*). Combined aliquots (200 mL) were centrifuged through 30 kDa molecular weight cutoff (MWCO) filter, and finally concentrated approximately twenty fold using a 3 kDa MWCO filter. The purified peptide was frozen in 1 mL aliquots. Aliquots of purified peptide were thawed, diluted with 3 mL of buffer D (8 M GuHCl, 50 mM Tris/HCl pH 8.0) and concentrated to the final volume of ∼300 μL using 3 kDa MWCO concentrator. Concentrated samples were loaded on a Tricorn 10/300 column (packed with Superdex

75 gelfiltration resin), and eluted at 1 mL/min using buffer E (20 mM sodium phosphate pH 8.0, 200 µM EDTA). Collected fractions (2 mL) were diluted with 6 mL of buffer D, concentrated, and the chromatography was repeated. After repetitive purification the Abeta monomer peak was collected on ice. The amount of purified Abeta was calculated by integration of the chromatographic UV absorbance peak, using extinction coefficient $E_{280} = 1,490 \, M^{-1} \, cm^{-1}$.

## Aggregation of abeta

Freshly purified (within 30 min after gelfiltration) monomeric Abeta42 solution was supplemented with 50 µM ThT and diluted using buffer E containing 50 µM ThT to reach different concentrations (1–6 µM). For the inhibition experiments 216, 36, 6, 1, and 0.167 µM solutions of flavones in buffer D, containing 2% DMSO and 50 µM ThT were prepared. Monomeric 6 µM Abeta42 solution was mixed with these flavone derivative solutions (or with buffer E containing 2% DMSO as a control) in a 1:1 ratio. Each sample was divided into three 100 µL aliquots into wells of a 96 well non-binding half-area plate (Corning NBS$^{TM}$). Kinetics of aggregation was followed at 37 °C temperature and constant shaking (960 rpm) using Synergy H4 Hybrid Multi-Mode (Biotek, Winooski, Vermont, USA) microplate reader. The intensity of ThT fluorescence was measured through the bottom of the plate every 3 min using 440 nm excitation and 482 nm emission.

## Aggregation of alpha-synuclein

Alpha-synuclein was dissolved in 30 mM Tris-HCl buffer (pH 7.5), containing 0.05% sodium azide and 50 µM ThT. For the study of the concentration dependence of fibrillation, a range of alpha-synuclein concentrations between 1 and 5 mg/mL (69–345 µM) was used, and for inhibition studies 150 µM final concentration was used. For the inhibition experiments 450, 375, 300, 225, 150, and 75 µM solutions of flavones in 30 mM Tris-HCl buffer (pH 7.5), containing 0.05% sodium azide, 2.25% DMSO, and 50 µM ThT were prepared. The 300 µM alpha-synuclein solution was mixed with flavones in a 1:1 ratio. Each sample was divided into three 160 µL aliquots into wells of a 96 well plate (Fisherbrand, Waltham, Massachusetts, USA) and one ∼3 mm glass bead was added into each well. Kinetics of aggregation was followed at 60 °C temperature and constant shaking (960 rpm) using Synergy H4 Hybrid Multi-Mode microplate reader. The intensity of ThT fluorescence was measured through the bottom of the plate every 2 min using 440 nm excitation and 482 nm emission.

## Extraction of relative $t_{50}$ and $I_{max}$ values

Maximal fluorescence intensities and halftimes of aggregation were obtained by fitting experimental data using following sigmoidal equation:

$$I = I_{min} + \frac{(I_{max} - I_{min}) * x^n}{t_{50}^n + x^n}$$

where $I$ is the intensity of fluorescence, $x$ is time, $t_{50}$ is the time to 50% of maximal fluorescence intensity, $I_{min}$ and $I_{max}$ are the minimal and maximal intensity of ThT fluorescence and $n$ is the Hill coefficient. Fitting was performed using Origin 8.1 software.

Extracted $t_{50}$ and $I_{max}$ values in the presence of flavone derivatives were divided by the values for the control samples to obtain relative $t_{50}$ and $I_{max}$. Average values and errors were calculated using three different batch preparations of flavones and three samples within each batch (a total of 9 repeats per flavone derivative).

## Congo Red spectroscopic assay

The assay was performed as described previously (*Nilsson, 2004*). Briefly, 7 mg/mL Congo Red stock solution was prepared in 30 mM Tris buffer, pH 7.5, and filtered through 0.22 μm syringe filter and stored at room temperature for up to a week. Alpha-synuclein samples were taken at different times of aggregation. 50 μL of alpha-synuclein sample was mixed with 50 μL of 70 μg/mL Congo Red (freshly prepared from stock solution) and incubated for 30 min at room temperature in 96 well non-binding half-area plate (Corning NBSTM). Spectra were recorded using Synergy H4 Hybrid Multi-Mode (Biotek) microplate reader. In case of Abeta, 35 μg/mL of Congo Red were added to the 15 μM peptide samples at the beginning of the reaction. 100 μL aliquots were transferred into wells of a 96 well non-binding half-area plate (Corning NBS; Corning Inc., Corning, New York, USA). The plate was incubated at 37 °C temperature, and spectra were recorded at different time points using Synergy H4 Hybrid Multi-Mode (Biotek) microplate reader. To get differential spectra, the corresponding spectrum at zero time point was mathematically subtracted from the spectra at later time points.

## Atomic force microscopy (AFM)

For atomic force microscopy experiments, 20 μL of the sample and 10 μL of 1 M HCl (protein fibrils in neutral/basic buffer have net negative charge and poorly adsorb on the mica) were mixed on freshly cleaved mica and left to adsorb for 1 min, the sample was rinsed with several mL of water and dried gently using airflow. AFM images were recorded in the Tapping-in-Air mode at a drive frequency of approximately 300 kHz, using a Dimension Icon (Bruker, Santa Barbara, California, USA) scanning probe microscope system. Aluminium-coated silicon tips (RTESPA-300; Bruker, Billerica, Massachusetts, USA) from Bruker were used as a probe.

## RESULTS

The first step of our study was the optimization of kinetic assays of Abeta and alpha-synuclein. As the best flavone inhibitors increased $t_{50}$ of insulin aggregation up to 24 times (*Malisauskas et al., 2015*), we tried to find conditions, where $t_{50}$ values of Abeta and alpha-synuclein aggregation would not exceed several hours. A way to get fast and highly reproducible kinetics of Abeta42 aggregation was recently described by the Linse group (*Hellstrand et al., 2010*). We were able to use it with similar results (Fig. 2A). Alpha-synuclein aggregates slower than Abeta; however, the rate of aggregation can be increased using agitation, beads, and higher temperature (*Buell et al., 2014*). The dependence of $t_{50}$ values on the concentration of alpha-synuclein is shown in Fig. 2B. For further experiments, we chose the optimal concentrations of peptides (3 μM for Abeta and 150 μM for alpha-synuclein).

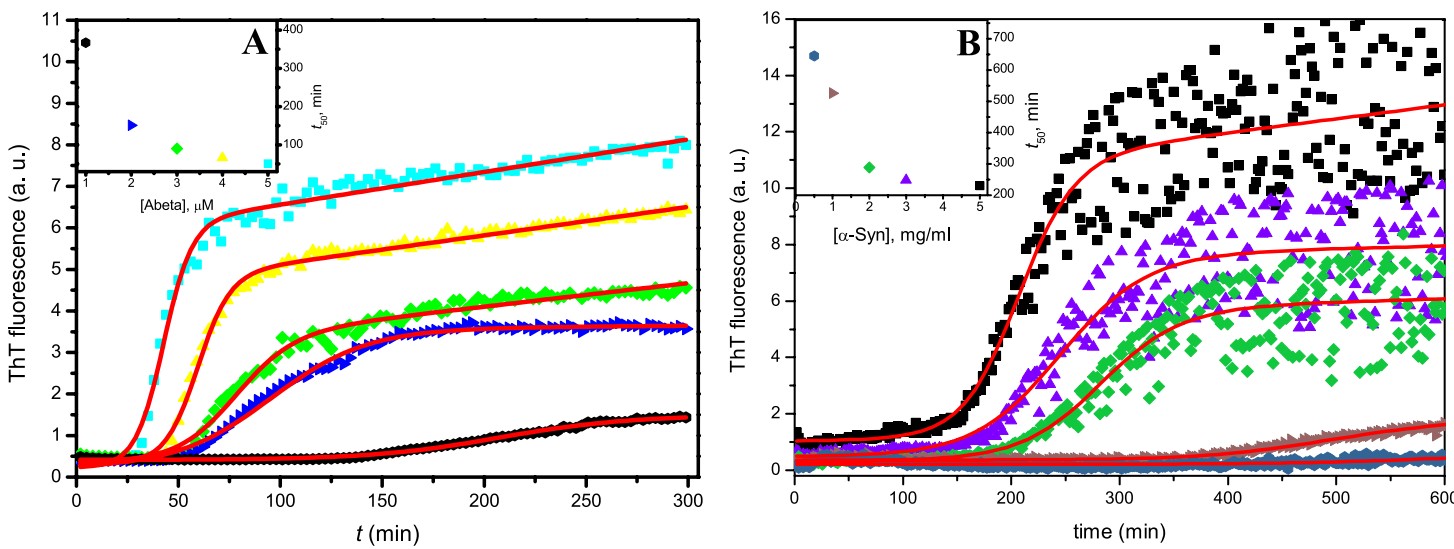

**Figure 2 Concentration dependence of Abeta (A), and alpha-synuclein (B) aggregation kinetics.** Raw data at different peptide concentrations is represented by scatter plots of different colors; fitting is represented by red curves. Inserts show concentration dependences of $t_{50}$ values and also serves as color-code legends.

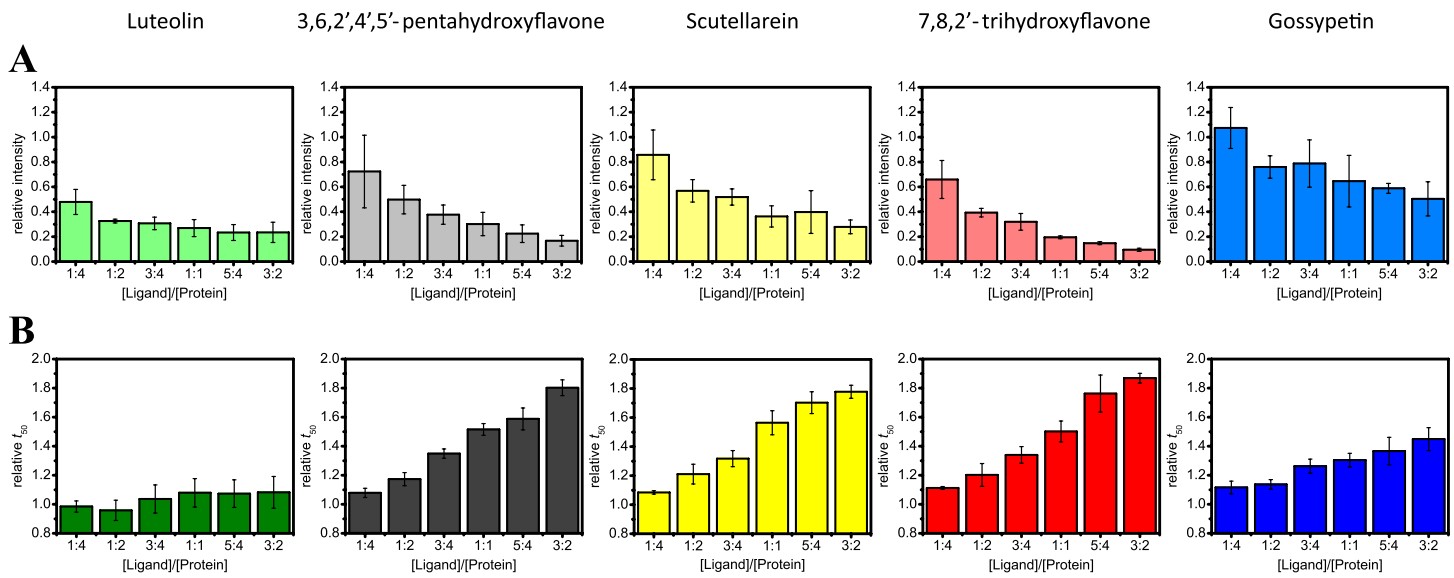

**Figure 3 Effect of flavones on the aggregation of alpha-synuclein.** Each flavone derivative is represented by different color. Relative ThT fluorescence intensities are shown in light colors (A) and relative halftimes of aggregation in dark colors (B).

The impact of flavones on ThT fluorescence intensity and $t_{50}$ values of alpha-synuclein aggregation is summarized in the Fig. 3. The effect of luteolin differs from other flavones. ThT fluorescence is strongly affected even by lowest luteolin concentrations, but relative $t_{50}$ values stay close to 1. Other flavones quench ThT fluorescence and increase the time of alpha-synuclein aggregation in a concentration-dependent manner. The inhibitory effect of gossypetin looks about twice lower than scutellarein, 7,8,2′-trihydroxyflavone or 3,6,2′,4′,5′-pentahydroxyflavone.

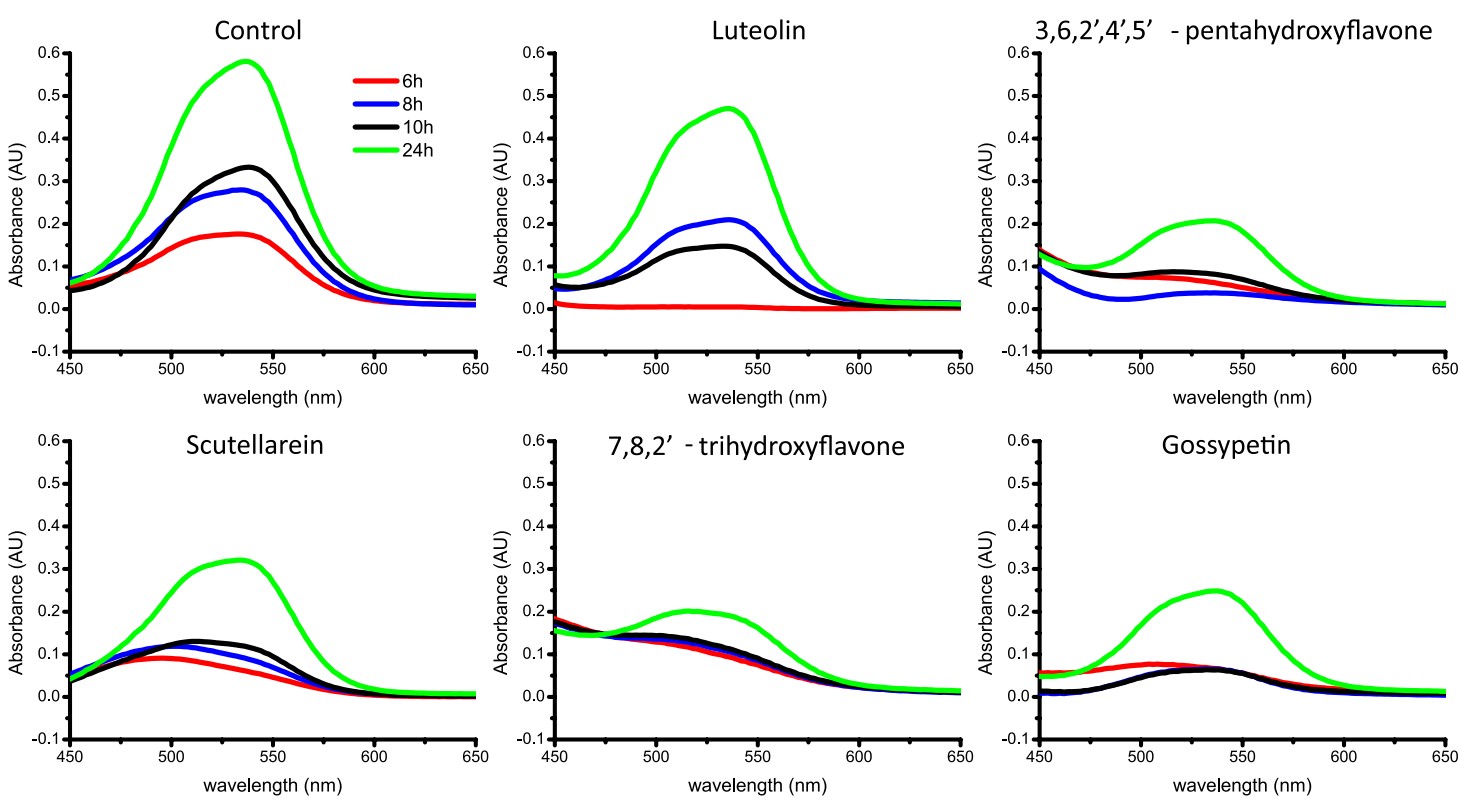

**Figure 4** **Aggregation of alpha-synuclein in the presence of flavones followed by Congo Red differential spectra.** The ligand/protein ratio was 3:2.

The complementary analysis using Congo Red spectroscopic assay gave similar results (Fig. 4). Spectral maxima at ∼540 nm show the fastest and highest increases in the control sample and in the presence of luteolin. In the presence of the rest of the flavones, there is just a minor rise within 10 h and smaller final absorbance after 24 h of incubation. The data suggest all tested flavones except luteolin inhibit aggregation of alpha-synuclein.

Microscopy data revealed that amyloid-like fibrils still can be formed in the presence of inhibitors (Fig. 5); however, some differences are worth mentioning. In the absence of flavones and in the presence of luteolin big fibril clumps can be found. In the presence of scutellarein and 7,8,2′-trihydroxyflavone, images contain mostly separate 4 nm diameter fibrils, while in the presence of 3,6, 2′,4′,5′-pentahydroxyflavone and gossypetin some fibril clumps and a number of 2–4 nm oligomers can be found. It suggests that all tested flavones but luteolin change not only time of aggregation but also the amount and the nature of final aggregates.

Equimolar amounts of flavones had almost no influence on both ThT fluorescence intensity and $t_{50}$ values of Abeta aggregation. In Fig. 6 we show the data, obtained using up to 36 times higher concentrations of compounds. Both fluorescence intensity and $t_{50}$ values of the aggregation in the presence of gossypetin show only minor decreases when compared to the control. In the presence of the highest used concentrations of other flavones, fluorescence intensity strongly decreases, but the time of aggregation is not increased. Moreover, the aggregation looks a little faster.

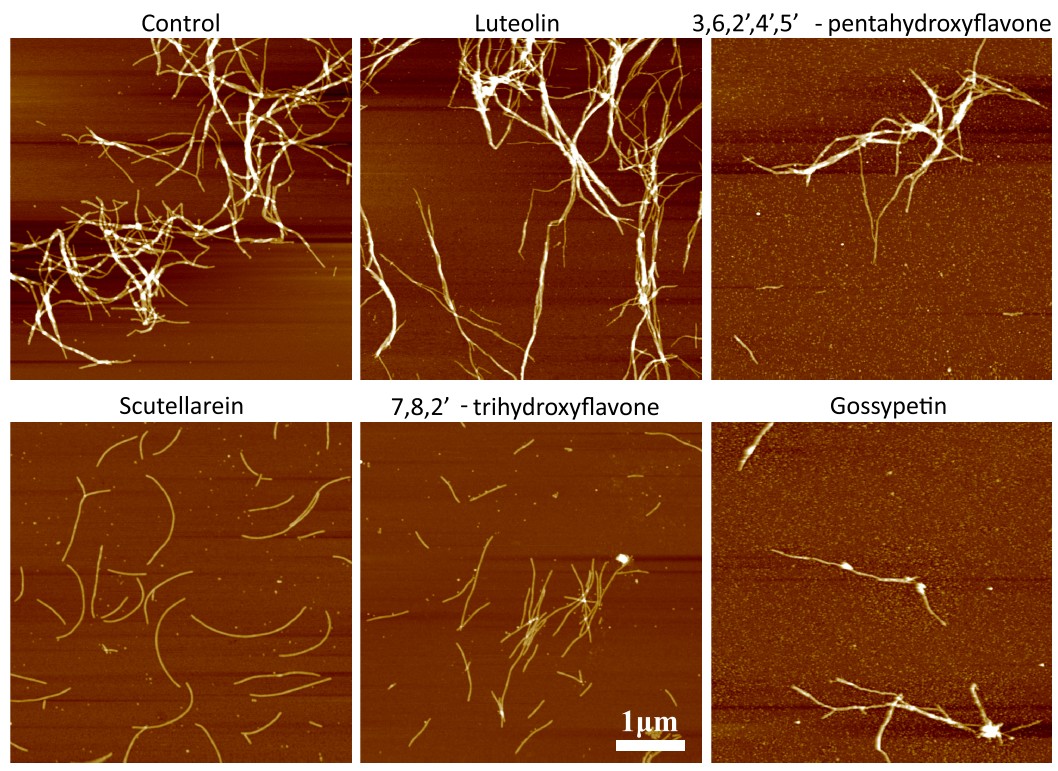

**Figure 5** AFM images of alpha synuclein aggregates. The ligand/protein ratio was 3:2.

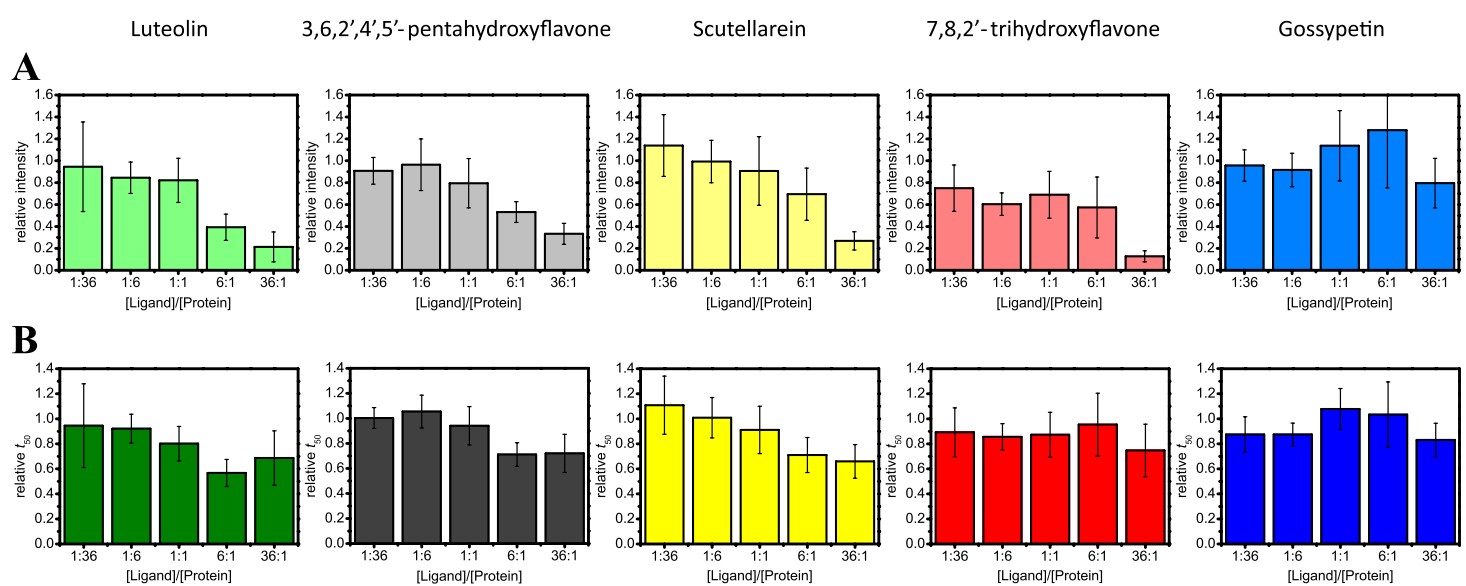

**Figure 6** Effect of flavones on the aggregation of Abeta. Each flavone derivative is represented by different color. Relative ThT fluorescence intensities are shown in light colors (A) and relative halftimes of aggregation in dark colors (B).

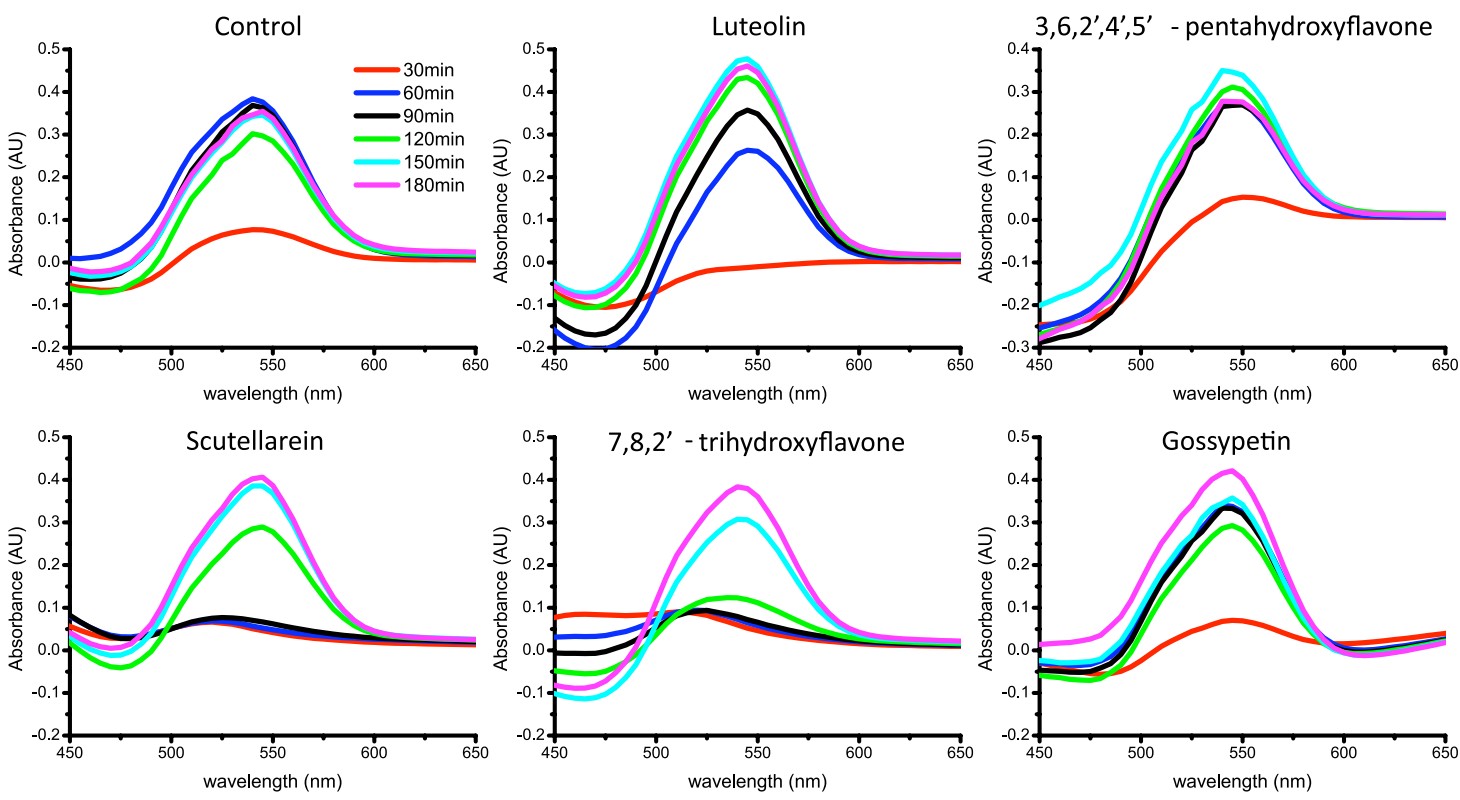

**Figure 7** **Aggregation of Abeta in the presence of flavones followed by Congo Red differential spectra.** Abeta concentration was 15 μM. The ligand/protein ratio was 7:1.

Complementary analysis using Congo Red spectroscopic assay showed some differences (Fig. 7). The spectral maxima at ∼540 nm grow fast in the control sample and in the presence of luteolin, 3,6, 2′,4′,5′-pentahydroxyflavone and gossypetin. In samples with scuttelarein and 7,8,2′-trihydroxyflavone the raise is stalled for the first 90 min; however, it reached similar maxima as in the control sample within 3 h. It can be interpreted as an inhibition of the initial steps of fibrillation; however the effect is not major.

No major differences were found in the AFM images of Abeta fibrils formed in the presence of different flavone derivatives (Fig. 8). The diameter of individual fibrils in all cases is in the range of 4–10 nm and the majority of fibrils are in larger clumps.

As a positive control of inhibition, we used EGCG (Fig. 9). In case of alpha-synuclein it works similar to most of the tested flavones. ThT fluorescence intensity is decreased and time of aggregation is increased with higher EGCG concentrations (Fig. 9A). Congo Red differential spectral maxima rise slowly and final absorbance values are low (Fig. 9B). Finally, a number of oligomers were detected together with the final fibrils (Fig. 9C). The only difference is higher ThT fluorescence intensities at low EGCG concentrations. In the case of Abeta, ThT intensity decrease and fluctuations of aggregation time are also comparable to the data seen in the presence of flavones (Fig. 9D). The Congo Red spectral maximum starts to rise after 2 h, similarly to the cases of scuttelarein and 7,8,2′-trihydroxyflavone (Fig. 9E). The diameter of Abeta fibrils formed in the presence of

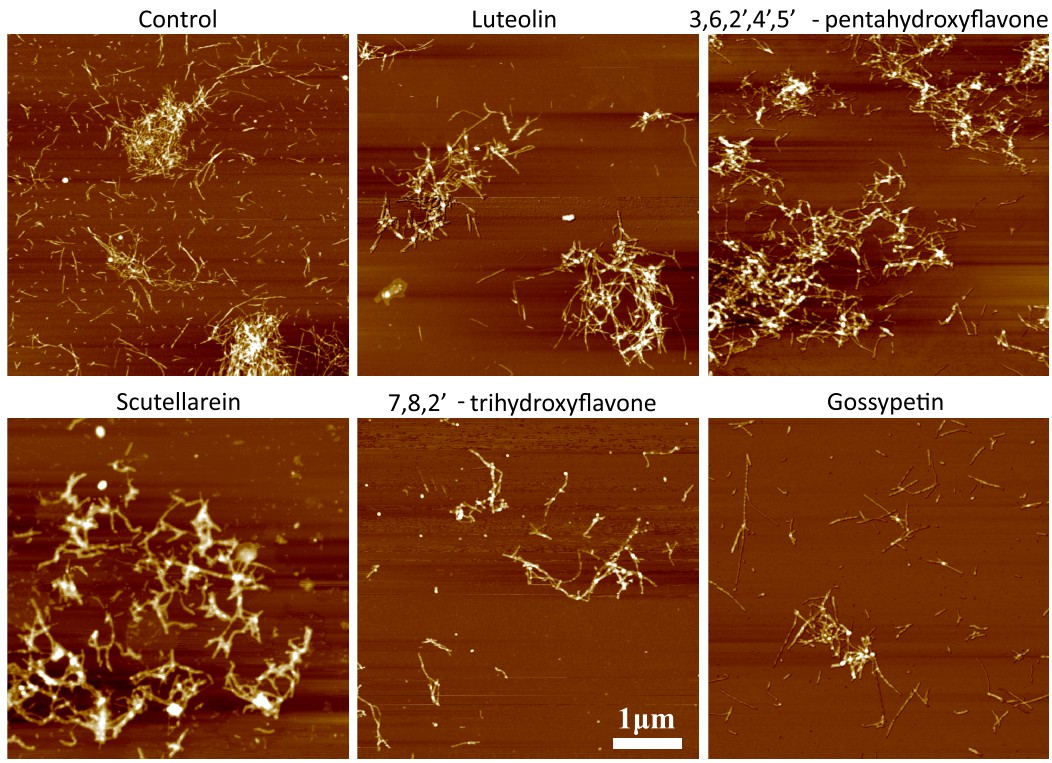

**Figure 8** **AFM images of Abeta aggregates. Abeta concentration was 15 μM.** The ligand/protein ratio was 7:1.

EGCG is also similar to the control sample; however, the amount of fibrils found on the mica is lower.

## DISCUSSION

We began the current study with hopes of finding a useful, generic inhibitor of amyloid fibril formation. With the variety of literature reporting flavones as effective inhibitors, we previously carried out an extensive screening of 265 commercially available flavone derivatives. The study revealed a method for eliminating large numbers of false positives for fibril inhibition while still using a simple method—ThT fluorescence. By measuring the time for 50% of maximal fibril growth rather than absolute ThT fluorescence, we identified only five flavones that appreciably inhibited insulin fibril formation (*Malisauskas et al., 2015*). In extending the study to look at the effects of these five flavones on Abeta and alpha-synuclein, we found moderate to no inhibition of fibril formation for either of these proteins. If there is an effective, generic inhibitor of fibril formation to be found among flavone derivatives, it is not among the 265 commercially available compounds.

Surprisingly, an expected positive control (EGCG) worked similar to some flavones. Our results show that EGCG does not inhibit Abeta fibrillation, leading to the conclusion that it is not the universal inhibitor implied by the collection of previous reports. In fact, our ThT data on aggregation of alpha-synuclein and Abeta in the presence of EGCG are comparable to the previously published (*Ehrnhoefer et al., 2008*) and the main controversy comes from the microscopy data. *Ehrnhoefer et al. (2008)* found just oligomers and no fibrils in

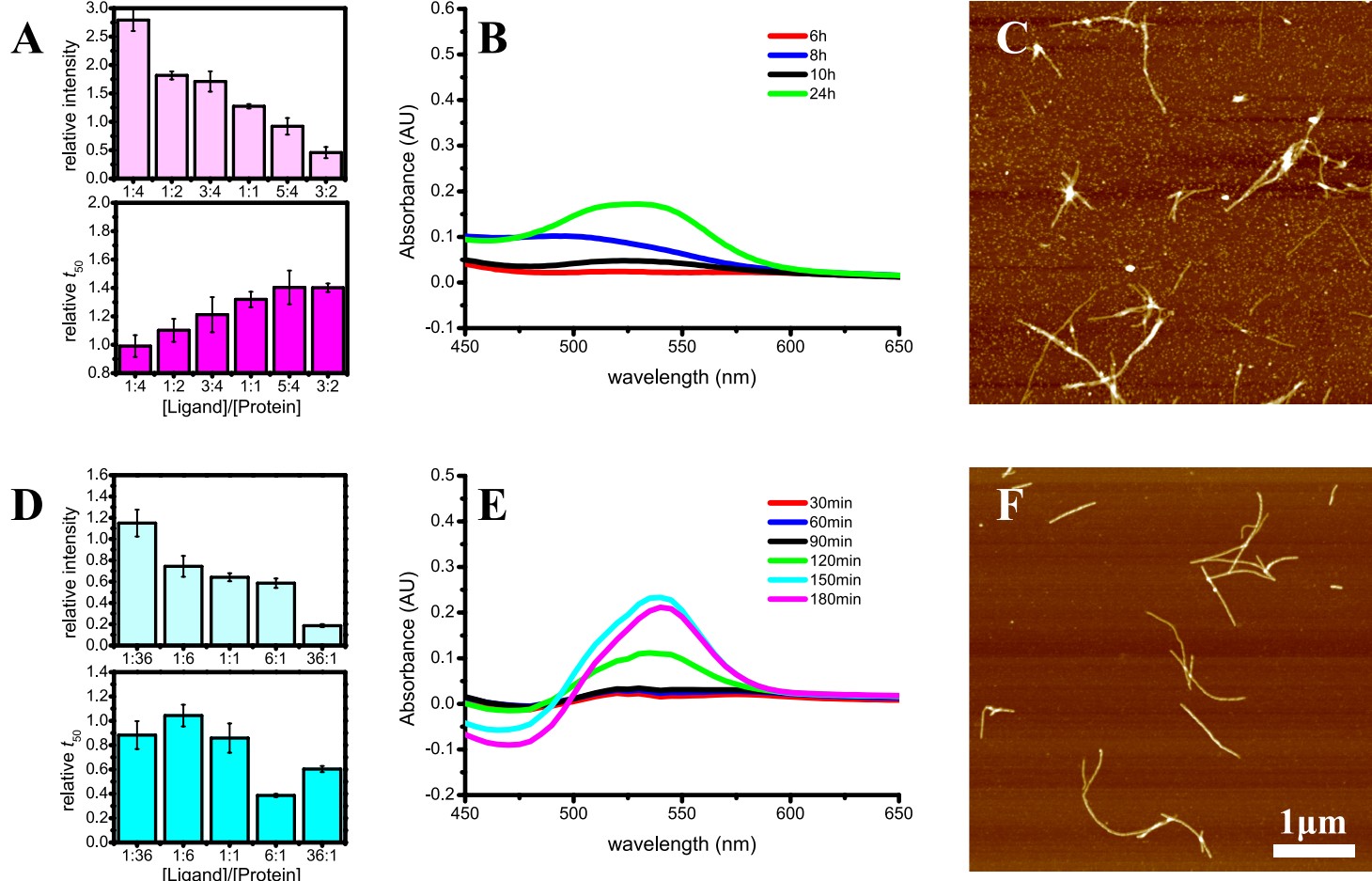

**Figure 9 Aggregation of alpha-synuclein (A–C) and Abeta (D–F) in presence of EGCG.** Illustrated by relative ThT intensity and $t_{50}$ (A and D), Congo Red differential spectra (B and E) and AFM images of final aggregates (C and F). Abeta concentration used for Congo Red and AFM studies was 15 μM. The EGCG/Abeta ratio was 7:1, and EGCG/alpha-synuclein ratio was 3:2.

alpha-synuclein and Abeta samples in the presence of EGCG; however, we can see mostly fibrils in Abeta samples (Fig. 9F) and fibrils together with oligomers in alpha-synuclein samples (Fig. 9C). Having in mind low stability of EGCG at neutral pH (*Zhu et al., 1997*) and the fact that EGCG effect on amyloid is dependent on auto-oxidation of EGCG (*Palhano et al., 2013*), slightly different conditions for our experiments (compared to Ehrnhoefer et al.) may be the reason of controversial data. If this is true, then even EGCG cannot be called universal inhibitor of amyloid aggregation, as inhibition may be dependent on the environmental conditions.

Failure appears to be a common state of affairs when discussing drug development for Alzheimer's disease (*Becker, Greig & Giacobini, 2008*; *Rosenblum, 2014*; *Schneider et al., 2014*). While failures in clinical trials involve much greater complexity than the (relatively) simple screening of small molecule amyloid inhibitors, many candidate molecules for drug trials come from high-throughput screening, so the method of screening is directly responsible for the selection of the right targets. As noted in the introduction and

confirmed in our current study, false positive results for amyloid inhibition can lead to extensive resources expended characterizing the effects of molecules that will ultimately be of no benefit in treating disease. Had we relied solely on maximum ThT fluorescence intensity to determine inhibition, four out of the five flavones studied would be indicated as inhibitors of Abeta aggregation. In fact, luteolin was already mentioned as an inhibitor previously (*Akaishi et al., 2008*). Of course, there is always thorough testing between initial screenings of drug candidates and clinical trials, but our simple expedient of measuring a kinetic feature of fibril aggregation has proven sufficient to eliminate many false positives with Abeta and alpha-synuclein fibrillation, in addition to our previous observations for insulin fibrillation.

Measuring the time to half maximal ThT fluorescence has proved so effective in the case of 265 flavones that we reduced a large false positive rate for inhibition to the point where we can demonstrate that there is no universal amyloid inhibitor among these candidate compounds. The more labor intensive, additional studies of measuring Congo Red absorbance spectra and AFM added some interesting information, but did not improve on our ability to quickly reduce a large set of candidate inhibitors to a small set of more promising candidates. However, care is clearly required before drawing sweeping conclusions from these simple results as illustrated by the discrepancies between our observations and those published previously regarding EGCG as an inhibitor of Abeta and alpha-synuclein fibrillation (*Ehrnhoefer et al., 2008*; *Bieschke et al., 2010*). These controversial data emphasize the ambiguity of methods applied for the detection of anti-amyloid compounds and suggests that differences in enviromental conditions must be carefully examined in seeking a full understanding of inhibitor effects.

## ACKNOWLEDGEMENTS

The expression plasmid (pRK 172) harbouring gene for human alpha-synuclein was kindly provided by Dr. LA Morozova-Roche and Dr. M Malisauskas. The expression plasmid with Abeta42 gene was kindly provided by Dr. C Cullin. The authors thank Dr. M Jankunec for the help with AFM.

### Funding

This research was funded by the European Social Fund under the Global Grant Measure, project number VP1-3.1-ŠMM-07-K-02-020. The funders had no role in study design, data collection and analysis, decision to publish, or preparation of the manuscript.

### Grant Disclosures

The following grant information was disclosed by the authors:
European Social Fund: VP1-3.1-ŠMM-07-K-02-020.

### Competing Interests

The authors declare there are no competing interests.

## Author Contributions

- Tomas Šneideris and Lina Baranauskienė conceived and designed the experiments, performed the experiments, analyzed the data, prepared figures and/or tables, reviewed drafts of the paper.
- Jonathan G. Cannon wrote the paper, reviewed drafts of the paper.
- Rasa Rutkienė and Rolandas Meškys contributed reagents/materials/analysis tools, reviewed drafts of the paper.
- Vytautas Smirnovas conceived and designed the experiments, analyzed the data, contributed reagents/materials/analysis tools, wrote the paper, prepared figures and/or tables, reviewed drafts of the paper.

## Supplemental Information

Supplemental information for this article can be found online at http://dx.doi.org/10.7717/peerj.1271#supplemental-information.

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
