# Peer review of "Looking for a generic inhibitor of amyloid-like fibril formation among flavone derivatives"

_PeerJ, doi:10.7717/peerj.1271_

## Round 0.1 · original submission · Major Revisions

· Academic Editor

Major Revisions

Particularly, you should add new experiments as suggested by Reviewer 1 and revise the introduction section as recommended by Reviewer 2.

Reviewer 1 ·

Basic reporting

No comments.

Experimental design

No comments.

Validity of the findings

In general, the manuscript meets all the standards cited in this session. However, I have questions about the employed methodology and suggestions that I believe will improve upon the quality of the work. Those are included in the General Comments for the Author box.

Additional comments

Although measuring the halftime of aggregation by thio-T binding is viable to follow amyloid formation, the negative/positive interference of Thio-T in the aggregation kinetics cannot be totally ruled out. Thio-T can compete with the compounds for binding either to protofibrils or mature ones. Besides, Thio-T can interact with the compound itself.

In my point of view, it is important to validate the described methodology using known anti-aggregating compounds (such as EGCG). As EGCG has been shown to be a generic inhibitor of amyloid fibril formation (as cited by the authors), using this compound as a positive control would be of value.

To prove that a compound is a true inhibitor of amyloid aggregation, the authors should do a complementary analysis, such as Congo red binding, binding to amyloid-specific antibodies, or TEM, etc. In my opinion, this is the only way how they can assure that the flavones (or any other compound) are truly inhibiting alpha-syn aggregation and are not inhibiting Abeta aggregation into ordered fibrils.

Abeta used in this study is produced by heterologous expression in bacteria. How do the authors guarantee that they are working with monomeric material? In general, authors work with Abeta1-42 produced by synthesis in solid phase.

Typos:

Lines 62-63. Correct to: “Recently, we demonstrated the ability to avoid false-positives in ThT fluorescence assay-based screening by comparing halftimes of aggregation…”

Reviewer 2 ·

Basic reporting

The manuscript is interesting yet the motivation, background and selection of compound should be better explained. It is not clear why flavones were selected as the lead compounds (aromatic interactions? other reasons?)

A QSAR analysis would be useful to make the manuscript more significant as many such works on screening on libraries are available

Experimental design

Fair

Validity of the findings

Fair

Additional comments

There are thousands of works on the inhibition of amyloid formation by small aromatic molecules. This should be presented either by citing more comprehensive review or better presentation in the "Introduction" section

---

## Round 0.2 · accepted · Accept

· Academic Editor

Accept

The authors properly addressed all the comments and requests of the reviewers.